# Photoacoustic and Magnetic Resonance Imaging of Hybrid Manganese Dioxide-Coated Ultra-Small NaGdF_4_ Nanoparticles for Spatiotemporal Modulation of Hypoxia in Head and Neck Cancer

**DOI:** 10.3390/cancers12113294

**Published:** 2020-11-06

**Authors:** Laurie J. Rich, Jossana A. Damasco, Julia C. Bulmahn, Hilliard L. Kutscher, Paras N. Prasad, Mukund Seshadri

**Affiliations:** 1Center for Oral Oncology, Roswell Park Comprehensive Cancer Center, Elm and Carlton Streets, Buffalo, NY 14263, USA; ljrich@buffalo.edu; 2Department of Chemistry and the Institute for Lasers, Photonics, and Biophotonics, University at Buffalo, The State University of New York, Buffalo, NY 14260, USA; jdamasco@mdanderson.org (J.A.D.); juliabul@buffalo.edu (J.C.B.); hlkutsch@buffalo.edu (H.L.K.); 3Department of Medicine, University at Buffalo, The State University of New York, Buffalo, NY 14260, USA; 4Department of Anesthesiology, University at Buffalo, The State University of New York, Buffalo, NY 14260, USA; 5Department of Dentistry and Maxillofacial Prosthetics, Roswell Park Comprehensive Cancer Center, Elm and Carlton Streets, Buffalo, NY 14263, USA

**Keywords:** head and neck cancer, tumor hypoxia, photoacoustic imaging, MRI, manganese dioxide nanoparticles

## Abstract

**Simple Summary:**

Tumor hypoxia is a documented negative prognostic factor that contributes to treatment resistance in head and neck cancer. In the present study, we use non-invasive magnetic resonance imaging (MRI) and photoacoustic imaging (PAI) to evaluate the ability of ultra-small manganese dioxide coated nanoparticles to modulate tumor oxygenation in vitro and in vivo. Our results highlight the utility of MRI and PAI in mapping tumor hypoxia and nanoparticle delivery and demonstrate the potential of image-guided nanodelivery in alleviating tumor hypoxia in head and neck cancer.

**Abstract:**

There is widespread interest in developing agents to modify tumor hypoxia in head and neck squamous cell carcinomas (HNSCC). Here, we report on the synthesis, characterization, and potential utility of ultra-small NaYF_4_:Nd^3+^/NaGdF_4_ nanocrystals coated with manganese dioxide (usNP-MnO_2_) for spatiotemporal modulation of hypoxia in HNSCC. Using a dual modality imaging approach, we first visualized the release of Mn^2+^ using T1-weighted magnetic resonance imaging (MRI) and modulation of oxygen saturation (%sO_2_) using photoacoustic imaging (PAI) in vascular channel phantoms. Combined MRI and PAI performed in patient-derived HNSCC xenografts following local and systemic delivery of the hybrid nanoparticles enabled mapping of intratumoral nanoparticle accumulation (based on T1 contrast enhancement) and improvement in tumor oxygenation (increased %sO_2_) within the tumor microenvironment. Our results demonstrate the potential of hybrid nanoparticles for the modulation of tumor hypoxia in head and neck cancer. Our findings also highlight the potential of combined MRI-PAI for simultaneous mapping nanoparticle delivery and oxygenation changes in tumors. Such imaging methods could be valuable in the precise selection of patients that are likely to benefit from hypoxia-modifying nanotherapies.

## 1. Introduction

Head and neck squamous cell carcinomas (HNSCC) are a unique group of neoplasms that exhibit aggressive biological behavior and therapeutic resistance [1]. In this regard, hypoxia is a major cause of therapeutic resistance and a documented negative prognostic factor in HNSCC [2,3,4,5]. As a result, there has been widespread interest in developing agents that modify tumor hypoxia (e.g., nitroimidazoles) [6]. While these agents offer the potential to enhance therapeutic efficacy (improved local control and survival), results from clinical studies of such agents in unselected patients have been disappointing, due to varying levels of oxygenation during disease progression and therapy [6,7,8]. Therefore, an improved strategy would be to utilize image-guided tumor hypoxia mapping to carefully select patients with hypoxic tumors that are likely to benefit from hypoxia-modifying agents [9].

Photoacoustic imaging (PAI) is a relatively new imaging technique that utilizes a combination of light and sound to enable functional imaging of the tumor microenvironment [10,11]. In PAI, pulsed laser light is used to excite endogenous chromophores (e.g., hemoglobin, melanin) in tissue that consequently undergo a thermoelastic expansion leading to the generation of pressure waves that are detected by ultrasound (US) transducers. Specifically, in the context of imaging hypoxia, studies by us and others have shown that differences in optical absorption characteristics between oxygenated hemoglobin (OxyHb) and deoxygenated hemoglobin (deoxyHb) can be exploited by PAI to obtain relative estimates of oxygen saturation (%sO_2_) in vivo [12,13,14]. The overall goal of the present study is to assess the potential of image-guided nanoparticle-mediated hypoxia modulation in HNSCC.

It is widely recognized that due to their high metabolic activity, tumor cells generate higher levels of reactive oxygen species (ROS) compared to normal cells [15,16]. Cancer cells are also known to take up glucose at a much higher rate compared to normal cells and convert it to lactate (Warburg effect) which leads to acidosis [15]. In this regard, MnO_2_ is known to catalyze the decomposition of H_2_O_2_ (aq) to H_2_O (l) and O_2_ (g), while the acidic tumor environment leads to the decomposition of MnO_2_ to Mn^2+^ [17]. To exploit this phenomenon, we developed ultra-small NaYF_4_:Nd^3+/^NaGdF_4_ nanocrystals coated with manganese dioxide (usNP-MnO_2_) that could generate oxygen in the presence of ROS in the tumor microenvironment. Here, we report on the synthesis, characterization, and potential utility of our hybrid usNP-MnO_2_ nanoparticles for spatiotemporal modulation of hypoxia in HNSCC.

## 2. Results

### 2.1. Synthesis and Characterization of usNP-MnO_2_ with Imaging Properties

To develop a nanoformulation with hypoxia modulating properties, ultra-small NaYF_4_:Nd^3+^/NaGdF_4_ nanocrystals (usNP) were aggregated together using a MnO_2_ shell to form hybrid nanoparticles (usNP-MnO_2_; Figure 1A). The MnO_2_ shell has multiple functional capabilities. First, it reacts with ROS, specifically, hydrogen peroxide (H_2_O_2_) present within the tumor microenvironment to convert it to H_2_O (l) and O_2_ (g). Second, it dissociates in the acidic environment of a tumor to produce Mn^2+^ and O_2_ (Figure 1B; Appendix A). Third, the Mn^2+^ generated is a paramagnetic ion and can therefore be detected by MRI on the basis of the enhancement of T1 contrast. Incorporation of Gd^3+^ ions in the nanocrystal shell (as NaGdF_4_) leads to the confinement of a large number of paramagnetic ions on the nanocrystal surface to provide strong MR contrast , thus enabling real-time tracking of these nanoparticles. Additionally, the disintegration of the MnO_2_ shell not only releases the paramagnetic Mn^2+^ but also exposes the Gd^3+^ ions incorporated in the ultrasmall nanoparticles, effectively enhancing the T1 relaxivity of the nanoparticles as a response. Given the MR imaging capability of our nanoparticles and the ability of PAI to non-invasively visualize and quantify oxygen saturation based on endogenous contrast (Hb), we employed a dual modality approach to visualize the release of Mn^2+^ (MRI) and map modulation of oxygenation (PAI) both in vitro, using vascular channel phantoms, and in vivo, using patient-derived xenograft models of head and neck cancer (Figure 1C).

The usNP core-shell structures were synthesized by generating monodisperse NaYF_4_:Nd^3+^ core particles through a diffusion-controlled process, followed by the epitaxial growth of a thin NaGdF_4_ layer, similar to procedures described previously [18,19], with a reduced reaction temperature for crystal growth to ensure the ultrasmall size. In this manner, high-quality usNP with a narrow size distribution (Figure 2A,B) and good crystallinity were readily synthesized. 

Coating the usNP with MnO_2_ resulted in usNP-MnO_2_ hybrid nanoparticles with a 1:3 molar ratio between Gd and Mn determined by inductively coupled plasma—optical emission spectrometry (ICP-OES) measurement and elemental analysis (EDX) (Appendix A). Bovine serum albumin (BSA) coating was added to improve the colloidal stability and reduce the non-specific binding of the usNP-MnO_2_ hybrid. Addition of the BSA on the surface of the nanoparticles was confirmed by Fourier-transform infrared spectroscopy (FTIR) (Appendix A) and resulted in a decreased zeta potential while the hydrodynamic size measured from dynamic light scattering (DLS) showed no significant change over a five-day period (Appendix A). Without BSA on the surface, aggregation and colloidal destabilization is easily observed in the usNP-MnO_2_ dispersion in H_2_O and dextrose 5% in water (D5W), as evidenced in the increased and fluctuating values of the hydrodynamic diameter. Dispersing the nanoparticles in cell culture media with serum also resulted in better stability, as proteins in the media can adsorb on the surface of the MnO_2_ to provide electrostatic repulsion. The zeta potential of the usNP-MnO_2_ nanoparticles significantly decreased from ~−4 meV to ~ −30 meV when placed in the cell culture media, however, this interaction was not enough, as all of the nanoparticles were also observed to settle to the bottom even after sonication (Appendix A). Some sedimentation at the bottom was also observed for the BSA-usNP-MnO_2_ in a time dependent manner, potentially due to the large size of the dispersed nanoparticles and desorption of BSA over time. Sonication or vortexing readily resuspended the material. The biocompatibility of the nanoparticles was evaluated by examining the viability of FaDu HNSCC cells in vitro (Appendix A). All materials used for in vitro and in vivo experiments were freshly prepared.

### 2.2. Oxygen-Generating Capacity of usNP-MnO_2_ In Vitro

The reduction of MnO_2_ to Mn^2+^ and O_2_ is known to occur through a reaction with H_2_O_2_ which is enhanced under acidic conditions [17], both of which are characteristic of the hypoxic tumor microenvironment. To visualize the activity of our usNP-MnO_2_ under reductive and acidic conditions in vitro, transmission electron microscopy (TEM) was performed on usNP-MnO_2_ before and after treatment with H_2_O_2_ (pH 6). Prior to H_2_O_2_ addition, TEM images reveal aggregated usNP-MnO_2_. Addition of H_2_O_2_ resulted in rapid decomposition and release of usNPs from aggregated usNP-MnO_2_. Next, we examined the oxygen generating capacity of H_2_O_2_ responsive nanoparticles in real-time using PAI. PAI was performed using blood containing vascular channels created within a tissue mimicking phantom as previously described [13]. Blood was collected from severe combined immunodeficient (SCID) mice via cardiac puncture during CO_2_ asphyxiation. Due to asphyxiation, the pre-treatment %sO_2_ levels of this blood was around ~40% (Figure 3A), which is at an optimal oxygen saturation level for hemoglobin to readily bind free oxygen. The usNP-MnO_2_ were added to blood at concentrations of 4, 20, 40, and 400 µM, in addition to one sample which received water only (0 µM). PAI was then used to determine blood %sO_2_ levels before and after H_2_O_2_ addition. Pseudo-colorized maps show that all samples were consistently low in %sO2 (Figure 3A, pre-H_2_O_2_) prior to H_2_O_2_. Following baseline imaging, 62.5 µM of H_2_O_2_ was added to reduce the usNP-MnO_2_ and drive the production of free oxygen in the samples. Post-H_2_O_2_ pseudo-colorized maps show a visual increase in blood %sO_2_ at all MnO_2_ concentrations (Figure 3A, post-H_2_O_2_). Estimates of blood %sO_2_ before and after H_2_O_2_ showed a linear relationship between increase in blood %sO_2_ following H_2_O_2_ and usNP-MnO_2_ nanoparticle concentration (Figure 3B).

We then examined the potential of our redox-responsive usNP-MnO_2_ nanoparticles to serve as theranostic agents for simultaneous detection by MRI. MRI is a clinically relevant imaging technique that offers high spatial resolution with excellent soft tissue contrast. Using gadolinium (Gd) enhanced MRI, we have previously reported on the angiogenic heterogeneity in HNSCC xenografts [20]. However, commercially available Gd agents exhibit low relaxivity and pose the risk of toxicities in some patients [21]. To overcome these limitations, we incorporated Gd into the crystalline shell of usNP and examined changes in MR contrast following the reduction of usNP-MnO_2_. To test this, usNP-MnO_2_ were loaded into NMR tubes at increasing Gd concentrations and T1 relaxation rate (R1), measurements were acquired before and after treatment with H_2_O_2_ (Figure 3C,D). Prior to release, only slight differences in R1 were observed for all samples despite increasing usNP-MnO_2_ concentration (Figure 3C). However, immediately following the addition of H_2_O_2_, a clear concentration dependent increase in R1 (Figure 3D) was observed, as the MnO_2_ coating decomposed and the Mn^4+^ (in the MnO_2_) was reduced to free Mn^2+^ ions.

### 2.3. Combined PAI and MRI of Hypoxia Modulation and Nanodelivery in HNSCC Following Systemic Administration of usNP-MnO_2_

Building on our encouraging in vitro observations, we tested the dual potential of our usNP-MnO_2_ to generate oxygen and allow the tracking of nanoparticle delivery in vivo. We hypothesized that, due to the elevated levels of ROS and acidosis within hypoxic regions of the tumor, the accumulation of usNP-MnO_2_ should lead to improved oxygenation and release of usNP in these regions. To test our hypothesis, PAI (to examine modulation of hypoxia in vivo) along with co-registered MRI (to visualize nanoparticle distribution) was performed in SCID mice bearing a subcutaneous (s.c.) HNSCC patient-derived xenograft (PDX).

B-mode US with co-registered PAI was performed before and after injection (100 µL of 240 µM usNP-MnO_2_, i.v.) to monitor changes in total hemoglobin concentration (Hbt) and %sO_2_, respectively. T1-weighted MRI was also performed to visualize the intratumoral delivery of nanoparticles. The panel of images in Figure 4A represent HbT (left), %sO_2_ (middle) and normalized T1W intensity maps (right) of the tumor before and after systemic (i.v.) injection of usNP-MnO_2_. As expected, only slight changes were observed in tumor HbT levels on PAI following nanoparticle injection. In contrast, %sO_2_ maps show a visible increase in tumor %sO_2_ following injection. Importantly, some areas of the tumor which exhibited the lowest %sO_2_ levels prior to injection were also those which showed the greatest increase in %sO_2_ following usNP-MnO_2_ administration, indicating that usNP-MnO_2_ can successfully reoxygenate these hypoxic regions. Corresponding T1W intensity maps demonstrate the accumulation of released usNP in regions consistent with %sO_2_ images, providing a cross validation of usNP-MnO_2_ delivery and improved tumor oxygenation. Individual pixel values were extracted from each imaging dataset to create a histogram of the PAI and MRI signal intensity values of all pixels within the tumor. Histogram analysis showed a considerable shift toward higher %sO_2_ levels (Figure 4C), reflective of the local increase in oxygenation within the tumor, while HbT levels remained relatively consistent (Figure 4B). A noticeable shift in T1W intensity was also observed, though not as noticeable as changes in %sO_2_ (Figure 4D). Quantification PAI analysis revealed a significant (*p* < 0.05) increase in tumor %sO_2_ levels, with a relative percent increase of ~35% (Figure 4E) that was sustained over the course of several minutes (Appendix A). A significant increase in T1 enhancement was also observed for all tumors on MRI (*p* < 0.05; Figure 4F).

### 2.4. US-Guided Delivery of Nanoparticles for Localized Modulation of Tumor Hypoxia

Due to the spatial and temporal heterogeneity in tumor vasculature and blood flow, the delivery of usNP-MnO_2_ to all hypoxic regions of the tumor through systemic administration may not be possible. A direct injection of usNP-MnO_2_ into these regions may therefore allow for localized modulation of oxygenation. Consequently, we utilized a US-guided intratumoral (i.t.) injection procedure for the spatially targeted administration of usNP-MnO_2_ (Figure 5).

Prior to injection, B-mode US and T2-weighted (T2W) MRI were performed to detect avascular/necrotic areas of individual tumors. As shown in Figure 5A, both US and MRI successfully detected hypoxic/necrotic regions (outlined in yellow) within the tumor (outlined in white) which were subsequently targeted for US-guided delivery of usNP- MnO_2_. Significant enhancement in normalized T1W intensity was seen on MRI, reflective of usNP accumulation in the injected region (Figure 5A top). Corresponding spatially co-registered PAI showed increased %sO_2_ (Figure 5A bottom) in the avascular region following i.t. injection, confirming the localized delivery and release of O_2_. Consistent with these images, the quantification of all of tumors revealed significant increases in %sO_2_ (*p* < 0.05; Figure 5B) and normalized T1W intensity (*p* < 0.05; Figure 5B).

## 3. Discussion

There is considerable interest in developing MnO_2_-based nanotherapeutics to improve tumor oxygenation [22,23,24]. In this work, we designed MnO_2_ nanoparticles containing Gd (usNP-MnO_2_) to enable dual MR and PA imaging of the accumulation and degradation of these nanoparticles in the tumor microenvironment. We first tested the ability of our nanoparticles to improve oxygenation and MRI contrast in vitro. Treatment of usNP- MnO_2_ with H_2_O_2_ (pH 6) lead to the decomposition of the nanoparticles, resulting in the release of bare usNP, which yielded enhanced T1 relaxation rates. The quantification of T1 relaxivity of our nanoformulation revealed levels ~10 times greater than commercially available Gd-based contrast agents (data not shown). Furthermore, the H_2_O_2_ driven reduction of usNP- MnO_2_ within vascular channel blood phantoms produced increases in blood %sO_2_ levels linearly associated with nanoparticle concentration. These findings were then translated in vivo, where PAI detected rapid enhancement in %sO_2_ levels following the intravenous administration of usNP-MnO_2_ that were spatially co-registered with MRI. Importantly, only slight changes were observed in tumor HbT levels, confirming that these observations were due to a shift towards higher oxygenated hemoglobin levels and not in overall hemoglobin concentration. Finally, we showed that US-PAI can be utilized for image-guided intratumoral delivery of usNP-MnO_2_ for the targeted modulation of hypoxia in avascular/necrotic regions of the tumor. Simultaneous MRI allowed for validation of nanoparticle accumulation within the tumor following systemic or local delivery of nanoparticles, highlighting the strength in our dual modality approach. PAI provides non-invasive, real-time imaging of tumor oxygenation levels with the potential to yield spatial and temporal information on hypoxia modification. MRI, on the other hand, is a clinically relevant imaging technique that offers high spatial resolution with excellent soft tissue contrast and provides information on tumor morphology and vascular function [20,25]. Our dual modality approach not only enabled cross-validation of data but also provided complimentary information on tumor biology (MRI to visualize nanoparticle distribution within hypoxic regions) and response (PAI to examine modulation of hypoxia in vivo). The strength of this approach was highlighted in our HNSCC-PDX studies, where individual tumors exhibited a heterogeneous response to usNP-MnO_2_ administration, suggesting that some tumors are more primed for MnO_2_-based oxygen generation than others. As such, PAI and MRI could play a critical role in identifying tumors which are unresponsive to MnO_2_ nanotherapeutics, and therefore allow them to switch to alternative hypoxia modifying therapies. Given that hypoxia is a known negative prognostic factor in HNSCC, and its modification has been shown to benefit these patients, our findings not only support the use of usNP-MnO_2_ for enhancing tumor oxygenation but also show the utility of dual modal hypoxia mapping methods such as PAI-MRI to evaluate their efficacy. While these findings are promising, there are limitations to this work. To begin, %sO_2_ levels were only monitored over several minutes following usNP-MnO_2_ injection. It will be important to observe how usNP-MnO_2_ accumulate and modulate tumor oxygenation over longer periods to understand the kinetics and optimal schedule for hypoxia modification using these nanoparticles. Although not examined in this work, this optimized schedule would be important for evaluating the combination of usNP-MnO_2_ with radiation therapy and immunotherapy in future studies.

## 4. Materials and Methods

### 4.1. Generation of usNP-MnO_2_

To generate the bimodal core-shell structure, the NaYF_4_:Nd^3+^ core was synthesized by a modified high-temperature co-precipitation method. According to this method, 9 mL oleic acid and 15 mL octadecene were added to a 100 mL three-neck flask containing 0.97 mmol of YCl_3_·6H_2_O and 0.03 mmol of NdCl_3_·6H_2_O. The mixture was heated to 160 °C and maintained for 1 h under Ar gas with constant stirring and then cooled to room temperature. Ammonium fluoride (4 mmol) and 2.5 mmol sodium hydroxide (NaOH) were dissolved in 10 mL methanol (MeOH) by sonication and then added to the mixture and stirred for 30 min. The temperature was then increased to 100 °C and maintained for 30 min to evaporate MeOH. Once all the MeOH was removed, the mixture was heated at 260 °C for 10 min before cooling to room temperature. NaYF_4_:Nd^3+^ core was then collected by adding the excess amount of ethanol (EtOH) and centrifuged at 7000 rcf for 5 min. The collected precipitate was washed with ethanol and finally dispersed in 5 mL hexane. To coat the NaYF_4_:Nd^3+^ core with a thin shell of NaGdF_4_, the core was added to 0.1 mmol of Gd (CF_3_COO)_3_ and 0.2 mmol CF_3_COONa dissolved in 8 mL oleic acid and 12 mL octadecene in a 100 mL three-neck flask. The mixture was degassed under Ar for 30 min at 100 °C and heated at 260 °C for 10 min before cooling down to room temperature. The ultrasmall NaYF_4_:Nd^3+^/NaGdF_4_ (usNP) were collected by adding the excess amount of EtOH and centrifuged at 7000 rcf for 5 min.

To form usNP-MnO_2_, Mn (VII) ions from KMnO_4_ were reduced to form the MnO_2_ coating, which aggregated the usNP into larger nanoparticles. In this case, 0.5 mg/mL usNP solution was prepared by dispersing 1 mg of usNP in a 1:1 mixture of dichloromethane and dimethylsulfoxide. The resulting clear solution was sonicated for 5 min prior to the addition under sonication of 100 μL of 6 mM KMnO_4_. The resulting brown solution was centrifuged at 7000 rcf to collect the precipitate and washed twice with H_2_O before re-dispersing in 1 mL H_2_O. The usNP-MnO_2_ were subsequently coated with bovine serum albumin (BSA) to aid in the colloidal stability of the usNP-MnO_2_ and to reduce their non-specific binding. The usNP-MnO_2_ solution (1 mg/mL) was added to 2 mg of BSA and bath sonicated continuously for 15 min to form the BSA coated usNP-MnO_2_. The particles were again collected by centrifugation at 7000 rcf and washed three times to remove excess BSA and finally redispersed in 5% glucose solution (D5W).

### 4.2. Nanoparticle Characterization

The morphology, structure, and size of NPs were characterized by transmission electron microscopy (TEM), characterizations were carried out using a JEM-2010 microscope (JEOL USA, Inc., Peabody, MA, USA) at an acceleration voltage of 200 kV. Samples were prepared by drop casting 10–20 μL of each solution on a copper grid with carbon-coated formvar support and allowed to air dry overnight prior to measurements. The hydrodynamic diameter and ζ-potential measurements were performed using a 90Plus zeta sizer (Brookhaven Instruments, Holtsville, NY, USA). Inductively coupled plasma optical emission spectrometer (ICP-OES) analysis was performed using a Thermo Scientific iCAP 6000 (Thermo Fisher, Waltham, MA, USA) to evaluate NP composition. Samples were digested in concentrated nitric acid overnight and then diluted for measurement. Elemental analysis was performed on a Carl Zeiss AURIGA CrossBeam Focused Ion Beam Electron Microscope (Zeiss, Oberkochen, Germany) using energy-dispersive x-ray spectroscopy (EDX). Samples were prepared by drop casting 15 µL on a copper grid with carbon-coated formvar support and allowed to air dry overnight prior to measurements. The Fourier-transform infrared (FTIR) spectra were recorded using a Spectrum II FTIR spectrophotometer with a micro-Attenuated Total Reflectance (ATR) sampling accessory (Perkin Elmer, Waltham, MA, USA). Samples were placed on a 2-mm diamond window and spectra were recorded in the wavenumber range of 400–4000 cm^−1^ with a resolution of 4 cm^−1^. The force gauge reading was 50 units. ATR correction was applied.

### 4.3. Vascular Channel Phantoms

Preparation of the tissue-mimicking vascular channel phantom was performed as previously described [13]. Briefly, a saline based solution containing 3% agarose and 1.5% intralipid was heated, cooled slightly, and poured into a mold strung with hollow polyethylene tubing. After the mixture solidified, the polyethylene tubes were removed, leaving behind hollow channels allowing for injection of blood. For in vitro blood reoxygenation studies, deoxygenated blood was collected from mice via a terminal cardiac puncture procedure following CO_2_ asphyxiation. usNP-MnO_2_ were then added to the blood at MnO_2_ concentrations of 0, 4, 20, 40, and 400 µM and these solutions injected into the phantom. PAI was then performed before and immediately following the addition of 62.5 µM H_2_O_2_.

### 4.4. Animal Studies

Experimental studies were performed using 8–12-week-old female SCID mice (C.B-Igh-1b/IcrTac-Prkdcscid) bred by the Laboratory Animal Shared Resource at Roswell Park Comprehensive Cancer Center. Mice were housed in sterile microisolator cages located within the LASR facility and provided standard chow/water in a HEPA-filtered environment and daily 12-hr light/dark cycles. All experimental procedures performed were reviewed and approved by the Institutional Animal Care and Use Committee (1183M; last approval date 08/06/2019; 938M; last approval date 04/03/2020). For usNP-MnO_2_ studies, animals (n = 5) were surgically implanted with a 2 × 2 mm tumor specimen obtained from a successfully established HNSCC patient derived xenograft (HNSCC-PDX) on the lateral flank. Tumors were allowed to grow to a volume of 350–500 mm^3^, and imaging was performed before and after intravenous (i.v.) and intratumoral (i.t.) injection of usNP-MnO_2_. The i.t. injection occurred one hour following i.v. administration. One tumor was excluded from the i.t. studies, due to the absence of necrosis within the tumor.

### 4.5. PAI with Co-Registered US

All US-PAI images were acquired using the Vevo 2100 (VevoLAZR, FujiFilm VisualSonics Inc., Toronto, ON, Canada) laser integrated high-frequency ultrasound (US) system. For in vitro phantom studies, a fresh phantom was placed on the imaging platform, stabilized, and injected with blood. For in vivo studies, animals were anesthetized with 2.5% isoflurane and secured to the heated platform. Hair was then removed over the tumor and gel applied to facilitate US transmission. All i.t. injections for usNP-MnO_2_ studies were performed by localizing the largest avascular/necrotic area within the tumor, and using the Vevo® image-guided precision micro-injection system (FujiFilm VisualSonics Inc., Toronto, ON, Canada) to efficiently deliver usNP-MnO2 (30 µL of 250 µM) to this region. The following PAI parameters were used for both in vitro and in vivo usNP-MnO_2_ studies; Transducer: LZ-250, Frequency: 21 MHz, Depth: 20.00 mm, Width: 23.04, Wavelength: 750/850 nm, Acquisition mode: sO2/Hbt. Single slice and 3D PA datasets were obtained before and immediately following both i.v. and i.t. injection of usNP-MnO_2_. Quantification of oxygen saturation (%sO_2_) and hemoglobin concentration (Hbt) levels in usNP-MnO_2_ studies were calculated through the VevoLab software (Version 1.7.2, FujiFilm Visualsonics, Toronto, ON, Canada) as previously described [6]. For phantom studies, %sO_2_ avg values, which exclude zero/void pixel areas and are representative of blood oxygenation, were obtained by tracing a circular ROI around each vascular channel for a single central slice of the phantom. For in vivo studies, %sO_2_ total measurements, which include zero/void pixels and are representative of tissue oxygenation, were obtained by tracing a 3D-ROI for the entire tumor.

### 4.6. Magnetic Resonance Imaging

Experimental MR imaging was performed using a 4.7T/33-cm horizontal bore magnet (GE NMR Instruments, Fremont, CA, USA) incorporating AVANCE digital electronics (Bruker Biospec with ParaVision 3.0.2; Bruker Medical Inc., Billerica, MA, USA). For in vitro phantom studies, usNP-MnO_2_ were added to NMR tubes at MnO_2_concentrations of 0, 15, 45, 90, 180, 300 µM inserted into a plastic honeycomb tube holder, and positioned inside the scanner. The temperature was maintained at 37 °C during the imaging procedure using an air heater (SA Instruments Inc., Stony Brook, NY, USA) and automatic temperature feedback system. T1 relaxation rates (R1) were measured using a Fast Imaging with Steady State Precession (True-FISP) imaging sequence with the following parameters: field of view (FOV) = 3.20 × 3.20 cm, matrix size = 128 × 128, TR/TEeff = 3.0/1.5 ms, NEX = 1, slice thickness = 1.5 mm, TI = 40.0 ms, flip angle = 60° and number of echos = 60. R1 maps were generated using MATLAB (Version 2015a; Mathworks, Natick, MA, USA), as previously reported, and R1 measurements were obtained using Analyze (Analyze PC, Version 7.0, Biomedical Imaging Resource, Mayo Clinic, Rochester, MN, USA) by tracing a circular region of interest (ROI) in the center of the phantom. For in vivo MRI studies, animals were anesthetized with 2.5% Isoflurane (Philips Medical Systems, Andover, MA, USA), secured onto a MR-compatible sled, and placed inside of the scanner. Image acquisition consisted of localizer scans, in addition to high-resolution T2-weighted (T2W) spin-echo and 3D T1-weighted (T1W) spoiled gradient recalled echo (SPGR) datasets. T1W 3D-SPGR images were acquired with the following parameters: matrix size = 192 × 128 × 128, field of view = 48 × 32 × 32 mm, TE = 3.0 milliseconds, TR = 15 milliseconds, flip angle = 25°, number of averages = 2. MR images were acquired prior to usNP-MnO_2_ injection, 10 minutes following intravenous administration, and 10 minutes following intratumoral injection. Following image acquisition, animals were removed from the scanner and monitored to ensure full recovery. Normalized T1-signal intensity measurements were obtained using Analyze (Analyze PC, Version 7.0, Biomedical Imaging Resource, Mayo Clinic, Rochester, MN, USA) by tracing an ROI around tumor and fat for each animal and dividing the fat signal from tumor signal. Pseudo-colorized R1 maps and T1-signal intensity images were generated using ImageJ (Version 1.47, National Institutes of Health, Bethesda, MD, USA).

### 4.7. Statistical Considerations

GraphPad Prism (GraphPad Software, Ver 6.0, La Jolla, CA, USA; www.graphpad.com) was used for all statistical analysis and graphical representation of data. For in vivo usNP-MnO_2_ studies, One-tailed paired Student’s *t*-tests were used to analyze differences in %sO_2_ total and normalized T1W signal intensity. Bars/points on graphs represent the mean, while error bars represent the standard error of the mean. *p* values of <0.05 were considered statistically significant.

## 5. Conclusions

Our studies highlight the usefulness of PAI and MRI to non-invasively visualize the modulation of tumor hypoxia in vivo. To the best of our knowledge, nanoparticle-enhanced PA and MR mapping of spatial and temporal heterogeneity of oxygenation and nanoparticle delivery in HNSCC has not been reported.

## Figures and Tables

**Figure 1 cancers-12-03294-f001:**
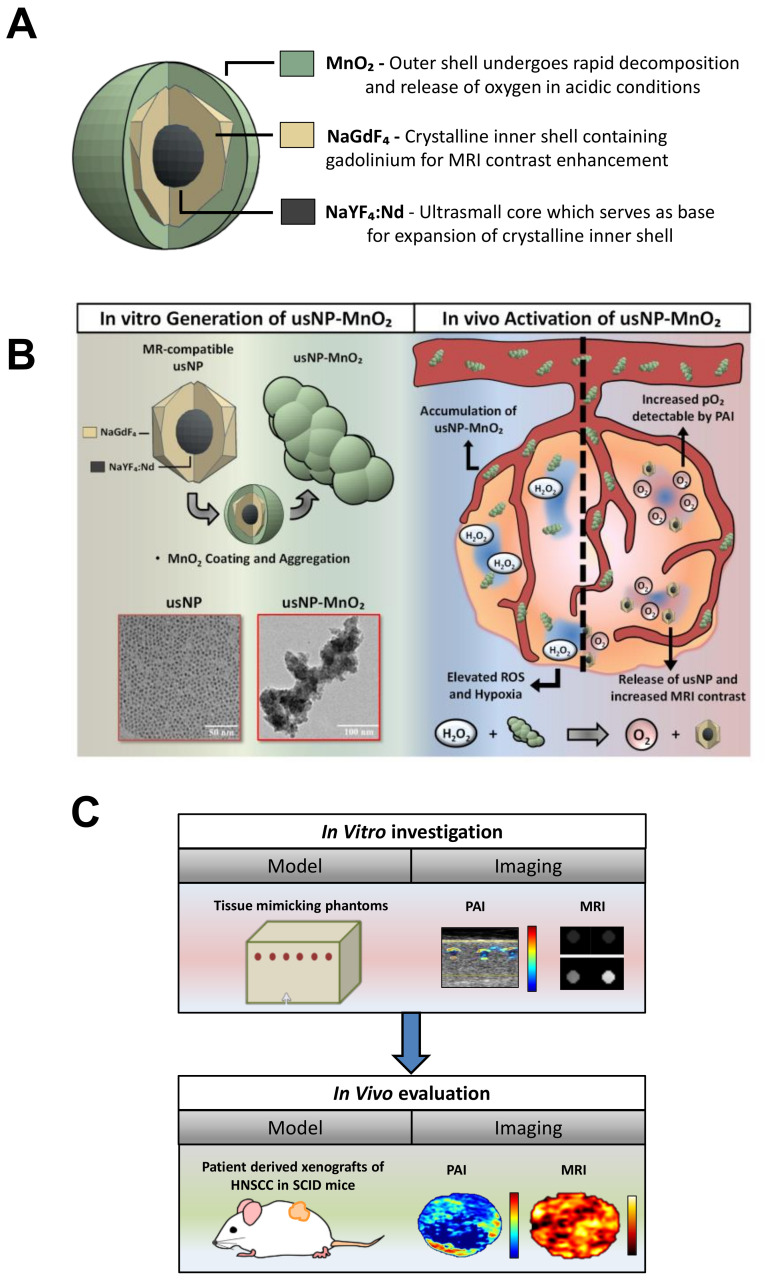
Photoacoustic and magnetic resonance (MR) imaging of hybrid manganese dioxide-coated ultra-small NaGdF4 nanoparticles for spatiotemporal modulation of hypoxia in head and neck cancer. (**A**) usNP-MnO_2_ composed of NaYF_4_:Nd^3+^/NaGdF_4_ nanocrystals coated with manganese dioxide were designed and synthesized for evaluation. (**B**) The hybrid nanoparticles have the potential to modulate tumor oxygenation by generating oxygen in the presence of reactive oxygen species (H_2_O_2_) within the tumor microenvironment. Areas of hypoxia (blue regions) within the tumor microenvironment are known to have lower pH and accumulation of reductive species such as H_2_O_2_. Under these conditions, usNP-MnO_2_ will accumulate and undergo a redox reaction to release O_2_ and free usNP, increasing pO_2_ detectable by PAI and allowing for MRI detection of usNP. (**C**) Photoacoustic and MR imaging was performed to assess the imaging properties and the ability of usNP-MnO_2_ to modulate hypoxia in vitro, using vascular channel phantoms, and in vivo, using patient derived xenograft models of head and neck squamous cell carcinoma (HNSCC).

**Figure 2 cancers-12-03294-f002:**
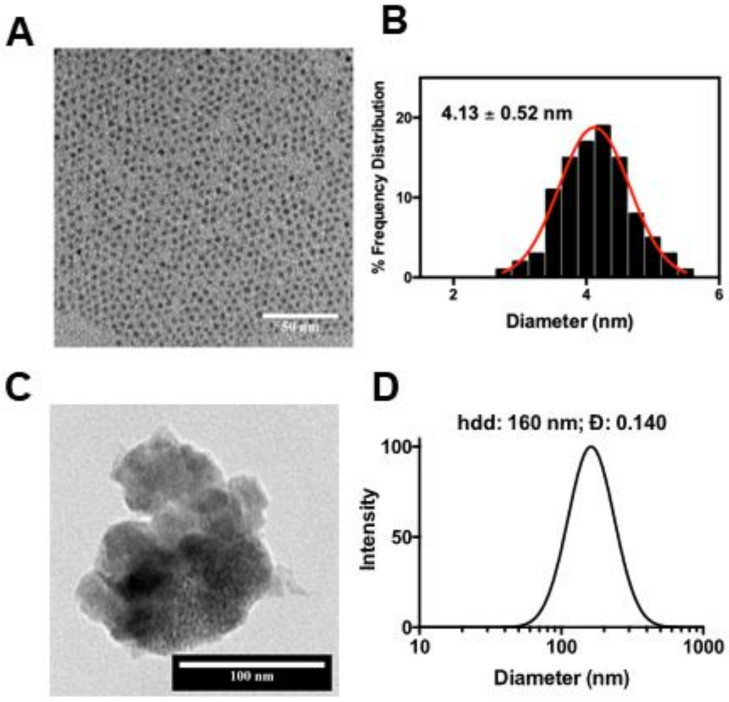
Size and morphology of nanoparticles. Transmission electron microscopy (TEM) image (**A**) of the synthesized usNP and the corresponding size distribution is shown (**B**). (**C**) TEM image of the synthesized BSA-usNP-MnO_2_ and the corresponding hydrodynamic size (**D**).

**Figure 3 cancers-12-03294-f003:**
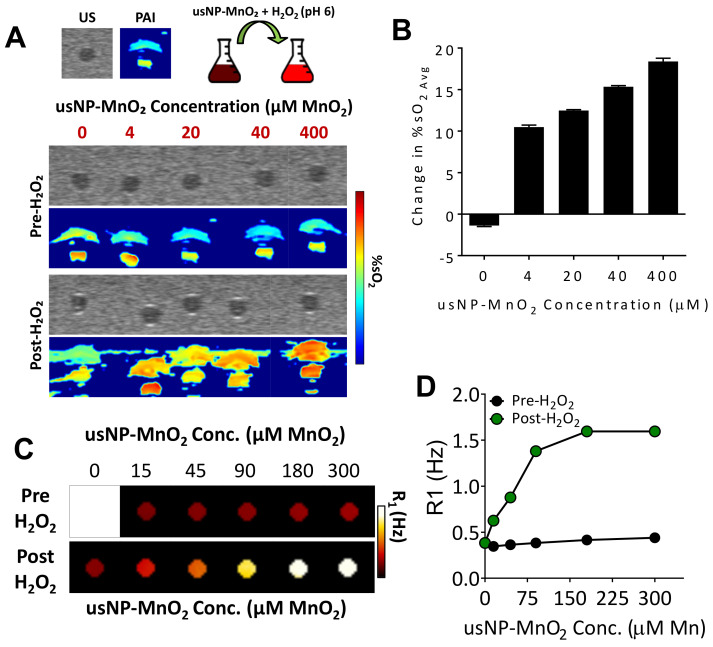
Oxygen-generating capacity and MR-compatibility of usNP-MnO2 under reductive and acidic conditions in vitro. (**A**) Pseudo-colorized oxygen saturation maps of blood containing varying concentrations of usNP-MnO_2_ before (pre) and after (post) the addition of H_2_O_2_. (**B**) Bar graph shows the change in blood %sO_2_ following the addition of H_2_O_2_ for each usNP-MnO_2_ concentration. (**C**) Pseudocolorized T1-relaxation rate (R1) maps of usNP-MnO_2_ in water at increasing concentrations before (pre) and after (post) exposure to H_2_O_2_. (**D**) Graph shows corresponding R1 measurements for each usNP-MnO_2_ concentration before and after H_2_O_2_.

**Figure 4 cancers-12-03294-f004:**
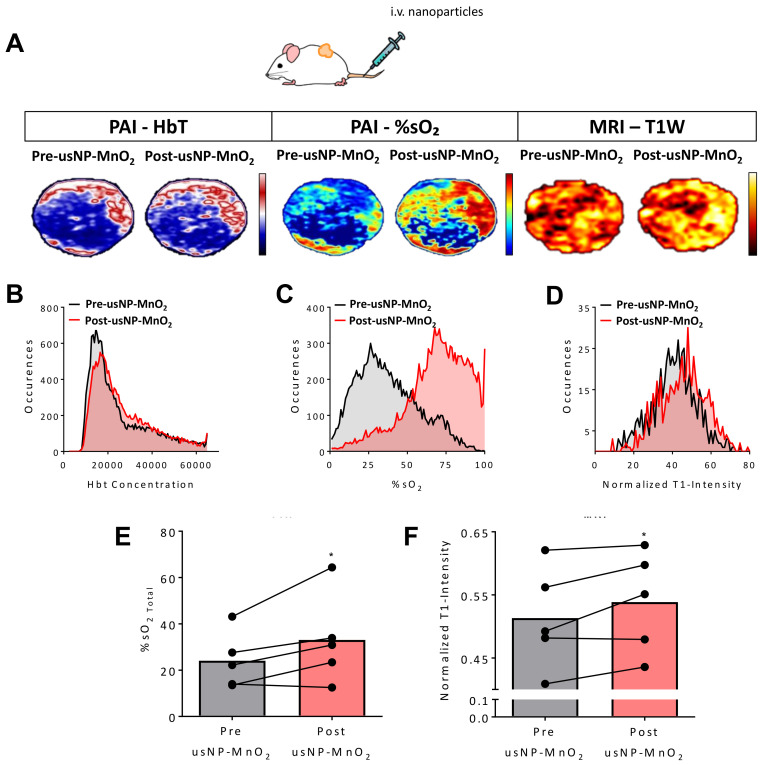
Modulation of hypoxia in head and neck squamous cell carcinomas (HNSCC) xenografts following systemic delivery of H_2_O_2_ responsive usNP-MnO_2_. (**A**) The panel of images represent co-registered Hbt (left), %sO_2_ (middle) and normalized T1-intensity maps (right) of an HNSCC patient-derived xenograft (PDX) before and after intravenous (i.v.) administration of H_2_O_2_ responsive usNP-MnO_2_. Corresponding histograms of hemoglobin concentration (Hbt; **B**), %sO_2_ (**C**), and T1-intensity (**D**) before and following usNP-MnO_2_ nanoparticle injection are shown. Graphs show increases in tumor %sO_2_ (**E**) and normalized T1W intensity (**F**) following nanoparticle injection, * indicates *p* < 0.05.

**Figure 5 cancers-12-03294-f005:**
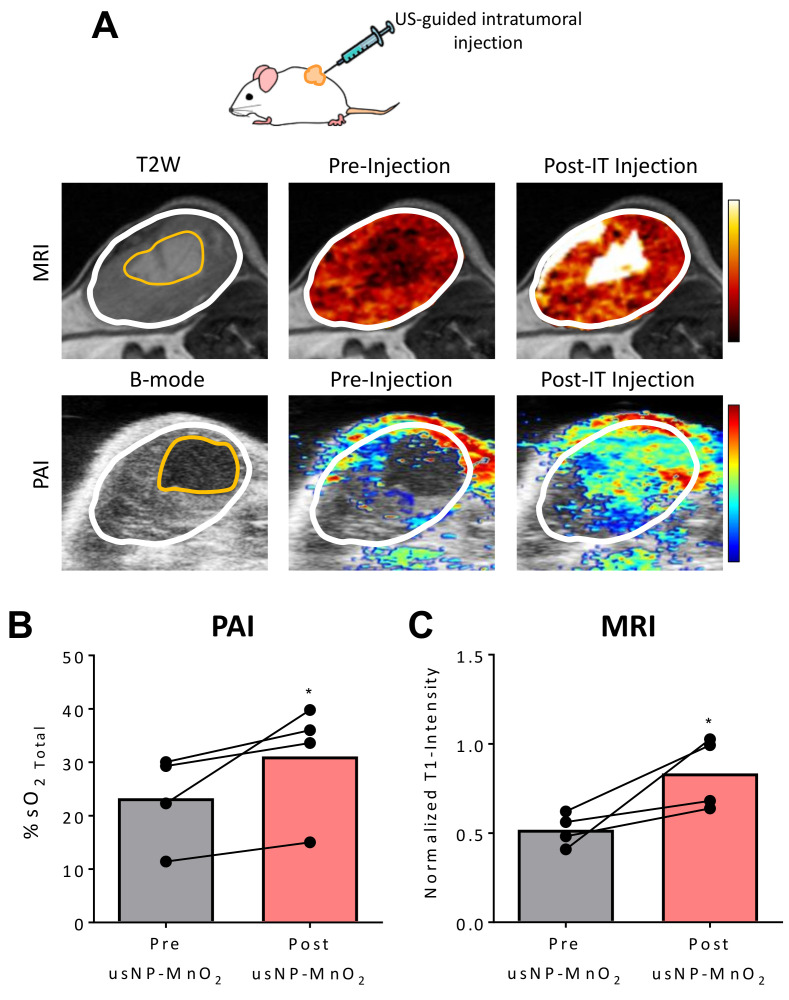
Ultrasound (US)-guided direct delivery of nanoparticles for localized modulation of hypoxia in the HNSCC microenvironment. (**A**) Panel of MR and photoacoustic (PA) images showing an HNSCC PDX before and after intratumoral (i.t.) injection of usNP-MnO_2_. US-enabled directed delivery of nanoparticles to avascular/hypoxic region (outlined in yellow) within the tumor (outlined in white). Graphs showing %sO_2_ (**B**) and normalized T1W signal intensity (**C**) before and after injection of usNP-MnO_2_, * *p* < 0.05.

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
