# Peer review of "Photoacoustic and Magnetic Resonance Imaging of Hybrid Manganese Dioxide-Coated Ultra-Small NaGdF4 Nanoparticles for Spatiotemporal Modulation of Hypoxia in Head and Neck Cancer"

_cancers, 2020, doi:10.3390/cancers12113294_

Round 1

Reviewer 1 Report

In this manuscript, the authors reported the preparation and the potential utility of ultra-small NaYF4:Nd3+/NaGdF4 nanocrystals with manganese dioxide (usNP-MnO2) for spatiotemporal modulation of hypoxia in HNSCC. Major corrections are needed before a publication in Cancers.

  1. The stability of usNP-MnO2 and BSA-usNP-MnO2 should be evaluated.
  2. The structure of usNP-MnO2 and BSA-usNP-MnO2 was only characterized by TEM. More characterization data is needed, such as XPS, FT-IR, EDS,..
  3. The authors claimed the usNP-MnO2 would reduce to Mn2+ in the presence of H2O2. Why is Mn2+, not Mn3+? More evidences should be provided to verify the generation of Mn2+ in vitro.
  4. The content of Mn ion in usNP-MnO2 should be calculated. Besides, in Fig. 3, the concentration of Mn should be given in vitro MRI test.
  5. The biocompatibility of usNP-MnO2 should be evaluated.

Author Response

In this manuscript, the authors reported the preparation and the potential utility of ultra-small NaYF4:Nd3+/NaGdF4 nanocrystals with manganese dioxide (usNP-MnO2) for spatiotemporal modulation of hypoxia in HNSCC. Major corrections are needed before a publication in Cancers.

  1. The stability of usNP-MnO2 and BSA-usNP-MnO2 should be evaluated.

Thank you for this comment. We have evaluated the stability of usNP-MnO2 and BSA-usNP-MnO2 by monitoring the aggregation over a 5-day period. Our data showed an improved stability with BSA coating, which correlated to an increase in the negative zeta potential (i.e. usNP-MnO2 has < -10, while BSA-usNP-MnO2 has ~ -30 values). Although some BSA-usNP-MnO2 has settled at the bottom over time due to the size of the usNP-MnO2 hybrid, the solution may be sonicated to resuspend the material and the measured hydrodynamic diameter did not change significantly. This data has now been included in the Appendix (Supplementary Fig. S4).

  1. The structure of usNP-MnO2 and BSA-usNP-MnO2 was only characterized by TEM. More characterization data is needed, such as XPS, FT-IR, EDS.

As suggested by the reviewer, we have performed EDX analysis to evaluate the elemental composition of the usNP-MnO2 and confirm the successful incorporation of all the elements in the synthesis. We have also performed FTIR analysis was also performed to verify the success of BSA coating. These results have been included in the Appendix (Supplementary Figs. S2 & S3).

  1. The authors claimed the usNP-MnO2 would reduce to Mn2+ in the presence of H2O2. Why is Mn2+, not Mn3+? More evidences should be provided to verify the generation of Mn2+ in vitro.

Although we did not perform experiments to verify the generation of Mn2+, it has been well studied that under acidic condition MnO2 can react with H2O2 to form Mn2+ along with H2O and O2. Mn3+ (MnOOH) is formed as an intermediate, which facilitates the decomposition of H2O2. However, it is further reduced to Mn2+, or maybe consumed to reform MnO2 as the reaction completes. A reaction scheme is included in the Appendix (Supplementary Fig. S1).

  1. The content of Mn ion in usNP-MnO2 should be calculated. Besides, in Fig. 3, the concentration of Mn should be given in vitro MRI test.

The Mn2+ concentration in usNP-MnO2 was quantified by ICP-OES. We have determined that there is a 1:3 molar ratio between the Gd and Mn in the usNP-MnO2 hybrid. This information has been included in the revised manuscript. The concentration in the vitro MRI test given in μM MnO2 concentration has equivalent μM Mn2+ concentration and has also been included in the revised version. The values in x-axis were changed to accurately reflect the concentrations of MnO2 μM concentrations.

  1. The biocompatibility of usNP-MnO2 should be evaluated.

We have now included results from a cell viability experiment conducted using the MTT (3-(4,5-dimethylthiazol-2-yl)-2,5-diphenyltetrazolium bromide) assay to evaluate the cytotoxicity of BSA-usNP-MnO2 in a FaDu head and neck squamous cell carcinoma cell line and determined the IC50 value at 90μM concentration in terms of MnO2 when incubated for 48h. This data has been included in the Appendix (Supplementary Fig. S5).

Reviewer 2 Report

Overview

The authors invented a special method to assess the tumor hypoxia by photoacoustic and magnetic resonance imaging using self-made ultra-small nanocrystals. The content of this manuscript seemed to be quite interesting to me and promising for future clinical application. It is well known by us, head and neck oncologist that the tumor hypoxia leads to the radio resistance of the tumor and the pre-treatment information of hypoxia of the tumor would help us to decide the treatment modality. Moreover, this material seemed to oxygenate the hypoxic area of the tumor, which would temporarily improve the radio sensitivity of the tumor.

Therefore, we need more information about the behavior of this material.

Major Points

#1. The authors mentioned about the limitation of this method in the discussion part.

“To begin, %sO2 levels were only monitored over several minutes following usNP- MnO2 injection”

Please show us the data of the time-decay curve of %O2 levels in the tumor.

It would be OK for the authors to give us details using supplementary tables or figures.

This data will help us to consider the future clinical application of this material.

Minor Points

P1L81-83:Abstract

“Given the MR imaging capability of 81 our nanoparticles and the ability of PAI to non-invasively visualize and quantify oxygen saturation 82 based on endogenous contrast (Hb), We employed a dual modality…..”

I could not catch up with the meaning of these sentences

P5L160

PDX

This should be "patients derived xenograft".

An abbreviation should be explained when it first appears in the manuscript.

Figure 4

I could not identify mark “C” or “D”.

Author Response

The authors invented a special method to assess the tumor hypoxia by photoacoustic and magnetic resonance imaging using self-made ultra-small nanocrystals. The content of this manuscript seemed to be quite interesting to me and promising for future clinical application. It is well known by us, head and neck oncologist that the tumor hypoxia leads to the radio resistance of the tumor and the pre-treatment information of hypoxia of the tumor would help us to decide the treatment modality. Moreover, this material seemed to oxygenate the hypoxic area of the tumor, which would temporarily improve the radio sensitivity of the tumor. Therefore, we need more information about the behavior of this material.

We are appreciative of the reviewer’s favorable review and thoughtful critiques.

  1. The authors mentioned about the limitation of this method in the discussion part. “To begin, %sO2 levels were only monitored over several minutes following usNP- MnO2 injection”. Please show us the data of the time-decay curve of %O2 levels in the tumor. It would be OK for the authors to give us details using supplementary tables or figures. This data will help us to consider the future clinical application of this material.

Thank you for this comment. As recommended by the reviewer, we have now included a dynamic data on the time-decay of %sO2 over several minutes post administration of our nanoparticles. %sO2 levels were found to increase rapidly and stayed elevated for several minutes (Appendix, Supplementary Fig. S6). A description of this figure has also been added to the manuscript (lines 201-202).

Minor Points

P1L81-83:Abstract

“Given the MR imaging capability of 81 our nanoparticles and the ability of PAI to non-invasively visualize and quantify oxygen saturation 82 based on endogenous contrast (Hb), We employed a dual modality…..” I could not catch up with the meaning of these sentences.

We have described the complementarity of our dual modality approach in the discussion section of the manuscript (lines 248-259 of the revised manuscript).

P5L160: PDX - This should be "patients derived xenograft". An abbreviation should be explained when it first appears in the manuscript.

We have revised the manuscript accordingly to address the comments.

Figure 4: I could not identify mark “C” or “D”.

Labels for Figure 4 have now been included.

Round 2

Reviewer 1 Report

As for stability test, please cite a reference (eg. Wu, F. S.; Yue, L. L.; Cheng, K.; Chen, J.; Wong, K. L.; Wong, W. K.; Zhu, X. J. Facile Preparation of Phthalocyanine-Based Nanodots for Photoacoustic Imaging and Photothermal Cancer Therapy In Vivo. ACS Biomater Sci Eng 2020, 6, 5230-5239.).

Overall, the paper is ready to be published in Cancer.

Author Response

As suggested by the reviewer, the reference has now been cited in the context of stability (Fig. S4). See attached files of the revised manuscript.

Reviewer 2 Report

I appreciate the authors’ response to my comments.

All but one point I raised were properly revised.

Please refer to minor point #3.

I still cannot find mark C or D in Figure 4.

Just put mark C and D on the left side of the histograms.

Author Response

Thank you. We apologize for the error. The labels "C" and "D" in Figure 4 have now been added.